# HIV-1 Persistence and Chronic Induction of Innate Immune Responses in Macrophages

**DOI:** 10.3390/v12070711

**Published:** 2020-06-30

**Authors:** Hisashi Akiyama, Suryaram Gummuluru

**Affiliations:** Department of Microbiology, Boston University School of Medicine, 72 E. Concord St., R512, Boston, MA 02118, USA

**Keywords:** HIV, macrophage, PAMPs, persistent viral RNA expression, chronic innate immune activation, intron-containing RNA

## Abstract

A hallmark of HIV-1 infection is chronic inflammation, which plays a significant role in disease pathogenesis. Acute HIV infection induces robust inflammatory responses, which are insufficient to prevent or eliminate virus in mucosal tissues. While establishment of viral set-point is coincident with downregulation of acute innate responses, systemic inflammatory responses persist during the course of chronic HIV infection. Since the introduction of combination antiviral therapy (cART), most HIV-1^+^ individuals can suppress viremia under detection levels for decades. However, chronic immune activation persists and has been postulated to cause HIV associated non-AIDS complications (HANA). Importantly, inflammatory cytokines and activation markers associated with macrophages are strongly and selectively correlated with the incidence of HIV-associated neurocognitive disorder (HAND), cardiovascular dysfunctions (CVD) and other HANA conditions. In this review, we discuss the roles of macrophages in facilitating viral persistence and contributing to generation of persistent inflammatory responses.

## 1. HIV Infection in Macrophages

Tissue-resident macrophages such as those in the lungs (alveolar macrophages), the central nervous system (CNS, microglia), the bones (osteoclasts) and the spleen (splenic macrophages) (reviewed in [1]) have been hypothesized to play an important role in HIV-1 pathogenesis [2]. Earlier studies in cART-naive individuals have shown that HIV-infected macrophages are frequently found in lymph nodes [3,4]. These studies reported on detection of HIV RNA and mature virions by immunohistochemistry and electron microscopy in lymphoid tissue-associated macrophages from HIV-1^+^ infected individuals with opportunistic infections. HIV-infected multinucleated macrophages have also been found in the brain from HIV^+^ individuals with encephalopathy [5]. Since it is difficult to obtain primary tissue-resident macrophages from humans, animal models of HIV infection such as SIV/SHIV infection of non-human primates (NHPs) and humanized mice have been widely used to study roles of macrophages in viral infection. Using NHP–SIV/SHIV models, it has been shown that macrophages play a predominant role as a source of viral replication and as a cause of tissue damage at late stages of disease progression when CD4^+^ T cells, the major target of HIV infection, are depleted [6,7,8]. A recent study in humanized mouse model of HIV-1 infection utilized electron tomography to reveal the presence of budding virions from infected bone-marrow resident macrophages [9]. In contrast to these studies, the role of macrophage infection in vivo has been challenged by the findings that transmitted/founder (T/F) viruses are unable to efficiently infect monocyte-derived macrophages (MDMs) [10,11,12,13]. The inability of T/F viruses, as well as some viruses isolated from chronically-infected patients, to establish MDM infections is correlated primarily to low cell surface CD4 expression in MDMs [11]. Interestingly, Calantone et al. claimed that SIV^+^ macrophages found in experimentally infected NHPs are not productively infected, but rather SIV antigen positivity might be attributed to phagocytosis of infected CD4^+^ T cells by macrophages [14]. While in vitro findings also suggest that macrophages can phagocytose or fuse with HIV-infected T cells [15,16], uptake of or fusion with infected T cells can lead to establishment of productive macrophage infection [15,16], suggesting that phagocytic or fusion-mediated delivery of infected cells can be an alternative route of HIV infection in macrophages [17,18]. These in vitro studies coupled with ex vivo findings that primary tissue-resident macrophages are susceptible to HIV infection [19,20,21] implicate that tissue-resident macrophages as virus reservoirs, regardless of the route of infection.

Additional support for macrophages as tissue reservoirs comes from studies of experimental SIV-infections of NHPs. For instance, macrophages isolated from various tissues of SIV-infected animals harbor replication competent viruses, as determined by the quantitative viral outgrowth assay (QVOA) [22,23]. Utilizing TCR beta as a marker of T cell contamination in macrophage preparations, these authors concluded that contribution of contaminated or phagocytosed infected T cells in the QVOA is negligible [22,23]. Macrophage tropic HIV-1 that can infect cells with low CD4 expression has been isolated from cerebrospinal fluid (CSF) of a patient on suppressive cART, suggesting production of HIV particles from replicating CNS reservoirs, that are most likely macrophages/microglia [24]. Persistent HIV infection of macrophages infection is further supported by the findings in a unique humanized mouse model of HIV-1 infection [25]. This mouse (myeloid-only mice, MoM) has engrafted human myeloid cells (monocytes, macrophages and dendritic cells) but is devoid of human T cells. Infection of MoM with HIV has resulted in persistent infection of human macrophages and, importantly, rebound of viremia after interruption of cART [25]. These studies suggest that while the route of infection and extent of virus production from infected macrophages would vary depending on anatomical locations and course of infection, tissue-resident macrophages contribute to HIV replication and persistence in vivo.

## 2. HIV Persistence in Tissue-Resident Macrophages

HIV-infected macrophages can serve as long-term tissue reservoirs of virus, particularly in the CNS. In experimental SIV infection of NHPs, SIV replication in brain macrophages could be detected as early as four days post virus inoculation [26]. Similarly, HIV infection can also be rapidly established in the brain soon after detection of peripheral viremia in blood within 15 days after infection with HIV [27]. Furthermore, HIV-infected macrophages can be found in patients, even on suppressive cART in diverse tissue sites (reviewed in [2,28]), although the mechanisms that account for long-term persistence of HIV^+^ macrophages remain unclear. While CD14^+^ macrophages such as dermal macrophages derived from circulating monocytes have a relatively short life (<6 days) [29], tissue-resident macrophages such as alveolar macrophages have a longer lifespan (>2 months) [30]. In contrast, microglia, the CNS-resident macrophages, are derived from yolk sac and are maintained in the brain for the entire life of an individual due to their self-renewal capacity [31,32,33,34,35]. In the human brain, 28% of microglia renew every year, and microglia age is about four years on average [36]. Thus, it remains formally possible that HIV-infected microglia might persist for the lifespan of the infected individual. In fact, recent QVOA studies optimized for myeloid cells demonstrated the presence of replication competent infectious viruses in microglia from SIV-infected cART-suppressed macaques [22,23,37]. Although macrophages have been postulated to harbor HIV-1 as transcriptionally silent provirus or unintegrated DNA (see the review in [38]), frequency and mode of HIV-1 latency in tissue-resident macrophages in patients on therapy are largely unknown. HIV latency has been hypothesized to be reactivated by multiple mechanisms such as co-infections and cytokines [39,40]. Whether latently infected microglia in HIV^+^ individuals are reactivated with such stimuli and to what extent HIV production from microglia can contribute to systemic viremia remain to be determined.

Macrophages in other anatomical locations are also persistently infected with HIV or SIV (reviewed in [2]). In the liver, HIV positive Kupffer cells have been found from HIV^+^ individuals on cART [41,42]. However, recovery of replication competent HIV ex vivo from liver-resident macrophages was unsuccessful [42]. Alveolar macrophages in the lung are HIV DNA/RNA positive [43], although whether infected alveolar macrophages can produce infectious progeny virions remains unclear. In SIV-infected NHPs on suppressive cART, tissue-resident macrophages from the lung and the spleen were viral DNA positive, expressed SIV gag RNA, and produced infectious particles upon stimulation [22,23,44]. Interestingly, urethral macrophages from SIV-infected monkeys express high levels of SIV RNA even during suppressive cART [45]. Recently, Ganor et al. reported that urethral macrophages from HIV^+^ patients on long-term (>3 years) cART harbor HIV DNA, RNA and proteins [46]. Importantly, HIV-infected urethral macrophages were induced to produce infectious particles upon ex-vivo stimulation with LPS, demonstrating urethral macrophages are one of the anatomical reservoirs of replication competent HIV in cART-suppressed patients [46]. Whether production of infectious HIV from these macrophages contributes to person-to-person transmission remains to be determined.

## 3. Pro-Inflammatory Responses in Tissue (CNS)-Resident Macrophages

A wide range of macrophage-associated immune activation markers have been linked to CVD [47,48], HAND, frailty [49], cancer [50] and pneumonia [51]. For example, elevated monocyte activation markers such as MCP-1, sCD14 and sCD163 have been repeatedly linked to CVD events or subclinical onset [52,53,54,55,56,57] and interestingly are independent predictors of all-cause mortality in virally suppressed cohorts [58,59,60]. Among tissue-resident macrophages, the contribution of HIV infected microglia to neuroinflammation has been most studied. Although cART has greatly reduced the severity of HAND, up to 50% of HIV^+^ individuals still suffer from neurocognitive disorders [61]. While the definitive cellular etiology of neurocognitive disorders in cART-suppressed HIV^+^ individual remains to be elucidated, chronic inflammation is postulated to be the chief driver of neuronal damage [61,62,63]. The primary virus positive cells in the CNS include brain perivascular macrophages and parenchymal microglia [64,65]. Microglia are unique among tissue-resident macrophages in that microglia can self-renew to maintain their population in the brain for the entire life [31,32,33] and are not re-populated by myeloid derived monocytes [31,32,33]. These unique features of microglia might shed light on the role of microglia in HIV persistence and neuroinflammation in the brain of HIV^+^ individuals on cART.

Although prolonged cART can suppress plasma viremia in HIV^+^ individuals under the detection limit for decades, viral RNA is detected in the CSF [66,67,68,69], suggestive of ongoing HIV transcription in CNS-resident cells including microglia and/or infiltrating cells. It has been reported that HIV infection of microglia can result in innate immune activation and neuronal injury (reviewed in [70,71,72]), although the molecular mechanisms underlying microglia activation and neuronal damage caused by HIV infection remain inconclusive. An obstacle to studying HIV infection in primary human microglia is the limited access to primary microglia, which restricts rigorous experimental strategies to elucidate molecular mechanisms of microglia infection and HIV-infection-induced immune activation. To overcome these limitations, many protocols have been developed to generate microglia from human induced pluripotent stem cells (iPSCs) [73,74,75,76,77,78]. iPSCs might provide an inexhaustible and reproducible source of cells, and they are amenable to gene-editing strategies such as CRISPR/CAS9 [79,80]. Additional advantages include the feasibility of establishing iPSC lines from patients with diverse genetic backgrounds, such as those with microglia-associated neurodegenerative diseases [81] or those with higher CNS HIV reservoirs, which might reveal pathways that could be harnessed to suppress disease states. A recent study has used commercially available iPSC-derived microglia and demonstrated that neuronal status (healthy or damaged) affects HIV replication in microglia, indicating interplay between microglia and neurons [40]. Further studies are warranted to use microglia and neurons from the same iPSC line and to include other CNS-resident cells such as astrocytes to form self-organizing three-dimensional organoid cell cultures to recapitulate cell-to-cell interactions and model brain structures in HIV-infected states.

## 4. HIV PAMPs and Induction of Persistent Immune Activation in Macrophages

Inflammatory markers associated with myeloid cell activation are strongly and selectively predictive of HAND and HANA [65,82,83,84]. Since access to tissue-resident macrophages is limited, MDMs have been used as a relevant in vitro model to study the role of macrophages in HIV-infection-induced inflammation. MDMs are equipped with multiple innate immune sensors that detect foreign pathogens to induce innate immune responses (Figure 1). Toll-like receptors are well-characterized sensors detecting viral and bacterial pathogens and detect HIV-associated viral RNA in some cell types (reviewed in [85,86]). However, HIV-mediated macrophage activation has been shown to be TLR-independent [87]. Decalf and colleagues demonstrated that HIV-1 fusion and entry into MDMs can trigger interferon-stimulated gene (ISG) expression in a TBK-1-dependent manner [88]. In contrast to HIV-1 entry, cytosolic DNA sensors (reviewed in [89]) have been hypothesized to sense the viral reverse transcription step. HIV RT products can be recognized by a cytosolic enzyme cGAS, which generates circular GMP-AMP dinucleotide (cGAMP), and cGAMP binds to STING to induce IFN-I responses via TBK1–IRF3 axis in MDMs [90]. Another study has shown that HIV RT intermediates trigger IFN-I production in MDMs in an IFI16–STING-dependent manner [91]. It should be noted that both studies utilized strategies to deliver SIV_mac_ Vpx via virus-like particles (VLPs) into the host cell cytoplasm to inactivate SAMHD1-mediated restriction of reverse transcription. SAMHD1, a pyrophosphatase that limits availability of dNTPs for reverse transcription in resting cells and terminally differentiated cells such as MDMs [92,93], is degraded by Vpx/Vpr alleles derived from several primate lentiviral lineages, including SIV_sm_/SIV_mac_/HIV-2 in a CUL4A dependent manner [94,95], thus facilitating robust reverse transcription and infection of non-dividing cells including dendritic cells and macrophages [92,93,96]. As a consequence of SAMHD1 restriction, the kinetics of HIV-1 RT reaction as well as the amount of HIV-1 RT products are limited in MDMs, when infections are initiated in the absence of SAMHD1 antagonism [97]. Not surprisingly, HIV-1 RT products in MDMs and dendritic cells in the absence of co-infection with SIV_mac_ Vpx containing VLPs fail to activate cGAS–STING-dependent innate immune sensing pathway [98,99,100,101,102].

It has long been appreciated that HIV-1 capsid remains associated with the viral cDNA and has been hypothesized to shield RT products from cytosolic nucleic acid sensing pathways (reviewed in [103,104]). In addition, host cells are equipped with cytosolic exonucleases, such as TREX1, that degrade excess amount of dsDNA in the cytosol to prevent constitutive activation of cytosolic DNA sensors [105]. Additional support for this hypothesis comes from recent findings that suggest disassembly of viral capsid occurs predominantly in the nucleus [106,107], thus further restricting cytosolic nucleic acid sensor(s) access to RT products and prevent host detection of the early steps of the viral life cycle. While cytosolic sequestration and shielding of viral nucleic acids by HIV-1 capsid is an attractive hypothesis, recent studies have described nuclear localization of cGAS and IFI16 [108,109,110], suggesting the existence of additional yet-to-be-defined viral mechanisms to disable nuclear-resident innate immune sensing mechanisms.

Other inducers of macrophage activation are viral proteins which are mostly characterized in microglia with regard to HIV-associated neurocognitive disorders. To date, HIV-1 proteins, Tat, gp120, Nef and Vpr have all been shown to activate microglia and affect their functions and neighboring neuronal health (reviewed in [72]). Although these findings are important, most of the studies have used overexpression of recombinant viral proteins or transgenic animals, and whether the concentration of these viral proteins used in these studies is physiologically relevant needs to be carefully considered. Johnson et al. detected Tat proteins by immunohistochemistry in infiltrating mononuclear cells, whereas HIV p24^Gag^ is not detected in brain biopsy samples from HIV^+^ individuals on suppressive cART [111]. In addition, Tat proteins were detectable by ELISA in three out of eight CSF samples from HIV^+^ individuals with undetectable plasma and CSF viremia [111]. Interestingly, one donor expressed over 30 ng/mL of Tat in the CSF, levels high enough to induce IL-6 secretion from fetal microglia in vitro [112]. Since the efficacy of certain cART regimens may be reduced in peripheral tissues such as the brain, allowing for the possibility of residual low-level viral replication and transcription [113], the role of viral proteins on chronic immune activation observed in cART-suppressed patients requires further analysis.

## 5. HIV Infection of Macrophages as a Driver of Chronic Inflammation

Many groups have shown that HIV-1 infection of MDMs induces pro-inflammatory cytokine production [98,114,115,116] and ISG expression [98,117,118,119], although robust IFN-I production has not been detected. In contrast, other studies have failed to observe robust innate immune activation in HIV-1 infected MDMs [99,100]. These differential findings may partially stem from differences in experimental setup such as differentiation protocols. For instance, supplementation of MDM culture media with M-CSF induces expression of phosphorylated SAMHD1 that does not have anti-viral activity [120,121], while addition of GM-CSF induces expression of G1/S-specific cyclin D2 and dephosphorylation of SAMHD1, thus limiting HIV infection [122]. Use of bovine serum instead of human serum is also known to alter activation status of MDMs and expression level of cell cycle-associated proteins including MCM2 and cyclins A, E and D1/D3 in M-CSF-differentiated macrophages [123]. Alternatively, the time post infection at which MDM activation was analyzed might also contribute to differences in reported infection-associated MDM activation outcomes. Recently, detailed quantitative analysis has revealed that completion of HIV reverse transcription and integration in MDMs takes 2–3 days post initiation of infection [97]. Therefore, it is highly likely that early stages of HIV replication (prior to integration) in MDMs are not subject to sensing by host nucleic sensors, but establishment of productive viral infection and especially late steps in the viral life cycle in MDMs induce pro-inflammatory responses.

## 6. Sensing of HIV RNA in Macrophages

Cells are equipped with numerous nucleic acid sensors to detect cytosolic viral RNAs. Well-characterized sensors include retinoic acid-inducible gene I (RIG-I)-like receptors (RLR) family members, RIG-I and MDA5 (reviewed in [124,125]), which recognize non-self single- and double-stranded RNA with an uncapped triphosphate group at the 5′ end and elicit robust IFN-I responses by inducing MAVS activation. While transfection of purified HIV RNA can induce RIG-I-dependent IFN-I responses [126,127], whether RIG-I can sense de novo transcribed HIV RNA upon infection remains unclear. We and others demonstrated that post-transcriptional HIV replication steps trigger innate immune responses in macrophages [98,117]. In particular, we showed that HIV RNA, and specifically intron-containing RNA (icRNA), induces MAVS-dependent IFN-I responses in MDMs, resulting in pro-inflammatory cytokine production and ISG upregulation (Figure 2) [98]. HIV RNA is transcribed as a ~9-kB fragment and undergoes splicing events similar to cellular mRNA, leading to synthesis of early viral protein products, Tat, Rev and Nef. Upon accumulation of Rev, icRNA (unspliced RNA or singly-spliced RNA) can be exported into cytosol in a Rev-RRE-CRM1-dependent manner (reviewed in [128]). Recent studies by us and others suggest that inhibition of HIV icRNA nuclear export alone attenuates induction of pro-inflammatory responses in HIV-infected MDMs [98] and dendritic cells [129]. Interestingly our studies also suggest that route of HIV icRNA nuclear export, specifically CRM-1 dependent pathway, is important for triggering cytosolic nucleic acid sensing mechanisms, since CTE (constitutive transporting element from Mason–Pfizer monkey virus [130])-dependent alternative nuclear export pathway (NXF1/NXT1-dependent pathway) failed to induce pro-inflammatory responses in MDMs. Importantly, cytosolic expression of HIV icRNA induced innate immune responses via MAVS, but MAVS activation was independent of RIG-I or MDA5 [98]. Increasing evidence suggests a non-redundant role for host RNA surveillance and degradation mechanisms such as nonsense-mediated decay and RNA exosomes in controlling foreign RNA and aberrant host-derived RNA (reviewed in [131]). Future studies are needed to elucidate molecular mechanisms of how host RNA surveillance mechanisms distinguish foreign RNAs such as HIV-1 icRNA from host-derived RNAs and identify the cytosolic RNA sensor that detects HIV icRNA in myeloid cells.

The current cART regimen in clinical practice includes entry, RT, integrase and protease inhibitors. Hence, once a provirus is established, none of the current ART can inhibit HIV proviral transcription or cytosolic HIV RNA export from nucleus. In fact, in HIV^+^ individuals or SIV-infected NHPs on suppressive cART, viral gag RNAs (i.e., icRNA) are still detectable in various tissues [46,132,133,134,135,136]. Although the majority of integrated proviruses are defective, containing large internal deletions and/or hyper mutations [137,138], these defective proviruses remain transcriptionally active and can lead to expression of icRNA [139,140]. It is conceivable that HIV icRNA from intact or defective proviruses are transcribed even in the presence of the current ART drugs, leading to chronic immune activation. Since tissue-resident macrophages are long-lived [30,31,32,33,34] and resistant to the cytopathic effects of HIV [141], continuous cytosolic HIV icRNA expression in tissue-resident macrophages may perpetuate chronic inflammation even in cART-suppressed HIV patients.

## 7. Pathological Consequences of Macrophage Inflammation by HIV-1 Infection

Macrophage activation and secretion of proinflammatory cytokines and chemokines, such as IP-10, caused by HIV infection might have a significant impact on HIV pathogenesis. For instance, IP-10 is one of the most abundant chemokines produced from HIV-infected macrophages [98,115]. IP-10 levels are highly associated with HIV disease progression, and IP-10 is known to suppress immune cell functions and facilitate HIV replication and dissemination (reviewed in [142]). IP-10 has also been shown to induce HIV latency in resting CD4^+^ T cells by altering actin structures [143], implying the role for activated macrophages in promoting HIV latency in secondary lymph nodes. Exposure of latently infected CD4^+^ T cells to immune activation stimuli can also result in spontaneous reactivation of viral transcription, which may lead to localized replication of HIV in tissues and contributing to blips of viremia. IFN-I is involved in activation of CD4^+^ T cell-derived HIV transcription [144], suggesting HIV-infected MDMs may contribute to viral transcription in latently infected CD4^+^ T cells in vitro. Moreover, we showed that HIV icRNA-expressing MDMs promote an immune exhaustion phenotype in co-cultured CD4+ and CD8+ T cells in an IFN-I-dependent manner [98]. In chronic HIV-1 infections, persistent immune activation can result in T cell exhaustion (reviewed in [145]). In particular, chronic exposure to IFN-I has been indicated as the driver of T cell activation [146,147], resulting in loss of immunological control of HIV-infected cells, thus contributing to HIV reactivation and persistence. Whether HIV icRNA expression in infected myeloid cells in vivo such as gut-resident macrophages in cART-suppressed animals or patients induce T cell exhaustion and promote virus persistence remains to be determined. It is also plausible that HIV-infection induced cytokines skew tissue-resident macrophages towards certain phenotypes [148]. Macrophages, while historically have been classified as either “pro-inflammatory” or “anti-inflammatory” cell subsets based on cytokines present in the culture conditions in vitro, are now known to be not mutually exclusive in function and are thought to exhibit diverse functional phenotypes [149]. Future studies are needed to define the diversity of functional phenotypes exhibited by HIV-infected tissue-resident macrophages and their contribution to tissue pathology.

The role of microglia activation caused by HIV infection in neuronal toxicity has been intensively studied, as described above. Microglia activation by HIV-1 infection induces pro-inflammatory cytokines which may directly affect neuronal viability, and viral proteins released from infected microglia may cause microglial activation, dysfunction and neuronal death (reviewed in [70,71,72]). Another important consequence of microglia activation by HIV infection is that it leads to dysfunction of microglia including reduced phagocytic activity [72]. Clearance of neurotoxins such as Tau proteins or fibrillar amyloid beta is an important homeostatic function of microglia, and persistently activated microglia have reduced capacity for phagocytosis of these potential neurotoxins [150]. It is plausible that HIV induced microglia activation leads to poor clearance of debris/neurotoxins that affects neuronal health. Elevated levels of activated microglia-derived neurotoxic metabolites, such as glutamate, arachidonic acid and quinolinic acid, have been reported in the CSF of HIV-1 infected individuals with HAND [151,152]. In addition, HIV^+^ individuals have a high risk of CVD including atherosclerosis. Markers of macrophage activation sCD14 and sCD163 are associated with progression of carotid plaque development [153]. Thus, it is likely that macrophages activated directly by infection or indirectly by cytokines increase risks of cardiovascular diseases in HIV^+^ individuals. Further studies on primary tissue-resident macrophages are warranted to investigate the role of macrophage infection and inflammatory responses on HIV-associated comorbidities.

## 8. Concluding Remarks

Resting CD4^+^ T cells are thought to be the major HIV reservoir in cART-suppressed patients. Recent findings have demonstrated that sequences of HIV-1 recovered from latently-infected resting CD4^+^ T cells in blood by in vitro stimulation are frequently different from those of rebound HIV-1 upon interruption of cART or latency reversing agent treatment [154,155,156]. These findings suggest the importance of studying HIV reservoirs in tissues. As discussed above, accumulating evidence suggests that tissue-resident macrophages are persistently infected, and thus it is highly plausible that these tissue macrophages also contribute to long-term viral reservoirs. In addition, pro-inflammatory cytokines produced from HIV infected tissue-resident macrophages may contribute to HIV pathogenesis and/or HANA conditions including HAND. Although our knowledge on the role of tissue-resident macrophages has been increasing, there remain numerous open questions. What are the molecular mechanisms that allow HIV persistence in tissue-resident myeloid cells? Do persistently infected macrophages contribute to systemic HIV dissemination? What strategies are needed to reduce viral reservoirs in tissue-resident macrophage reservoirs in addition to T cells? Does HIV infection in tissue-resident macrophages induce pro-inflammatory responses? What are the molecular mechanisms that contribute to myeloid cell-activation induced immunopathology? To date, most of the studies have utilized MDMs as a model of tissue-resident macrophages due to the limited access to these cell types from tissues. Although iPSC-derived microglia have been the focus of numerous recent studies as an in vitro model of CNS-resident macrophages [73,74,75,76,77,78], development of other tissue-resident macrophages from iPSC-lines, such as alveolar macrophages, Langerhans cells and Kupffer cells, has also been attempted (reviewed in [157]). Recent advances in generating tissue environment niches using iPSC-derived organoids [158,159,160,161] might provide a platform to terminally differentiate iPSC-derived macrophage progenitors to tissue-resident macrophages specialized in unique tissue environments [157]. Use of these stem cell-derived, self-organizing three-dimensional cell culture model systems might provide unique in vitro tools for performing rigorous studies on mechanisms of HIV persistence and chronic immune activation in tissue-resident macrophages, and provide unique insights into the role of macrophages in HIV-1 pathogenesis.

## Figures and Tables

**Figure 1 viruses-12-00711-f001:**
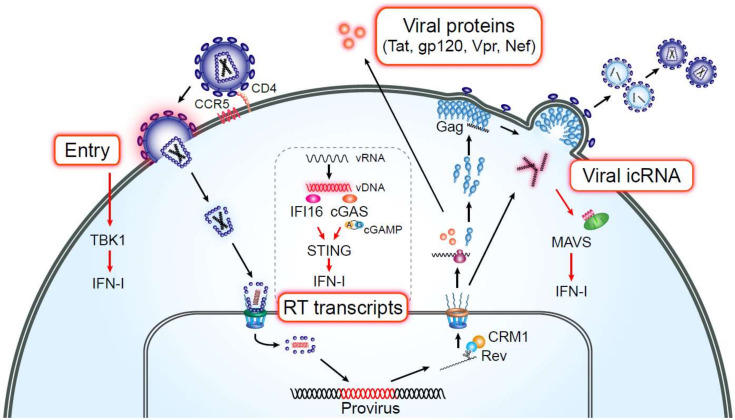
HIV PAMPs in macrophages. Multiple steps of the HIV-1 life cycle are detected by pathogen sensing mechanisms in macrophages. HIV-1 fusion and entry [88], over-exuberant expression of HIV-1 RT transcripts (upon SAMHD1 antagonism) [90,91] and de novo expression and Rev–CRM1-dependent nuclear export of HIV icRNA to the cytosol [98] can all lead to induction of ISG expression and IFN-I responses. Exposure to soluble viral proteins can activate tissue-resident macrophages, such as microglia (reviewed in [72]), resulting in secretion of pro-inflammatory responses.

**Figure 2 viruses-12-00711-f002:**
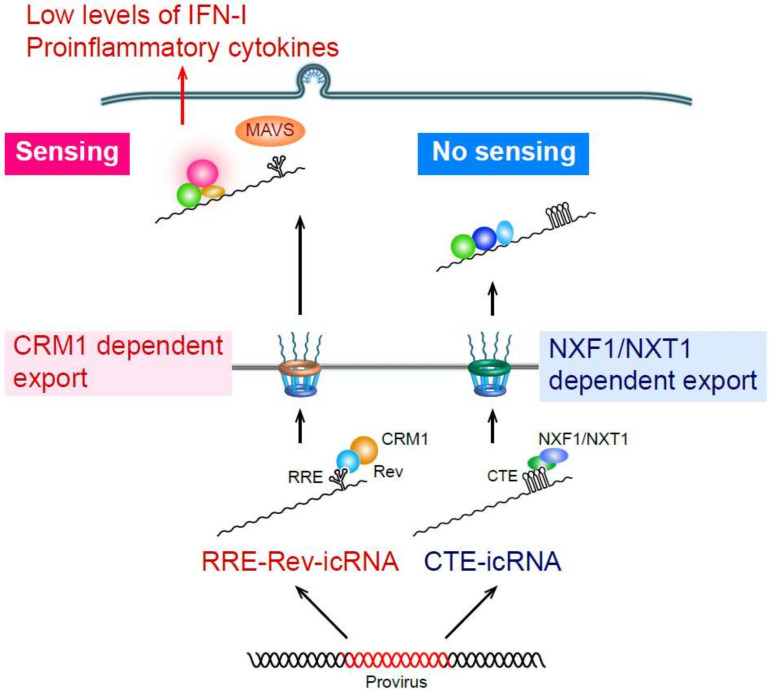
HIV icRNA-induced inflammation in macrophages. HIV icRNA exported from nucleus to cytosol by the Rev–CRM1-dependent pathway (RRE–Rev-icRNA) is sensed by a yet-to-be-identified RNA sensor, triggering ISG and IFN-I expression and pro-inflammatory cytokine production via MAVS. In contrast, cytosolic HIV icRNA exported by the NXF1/NXT1-dependent pathway (CTE-icRNA) does not result in ISG expression and IFN-I production.

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
