# Peer review of "HIV-1 Persistence and Chronic Induction of Innate Immune Responses in Macrophages"

_viruses, 2020, doi:10.3390/v12070711_

Round 1

Reviewer 1 Report

This review by H. Akiyama and S. Gummuluru deals with the roles of macrophages during the persistent inflammatory responses upon HIV-1 infection. The manuscript is very interesting as it reviews the available literature in the field to estimate how macrophages could contribute to the anti-viral immune response in tissues, mainly focusing on the brain. I found all the details extremely organized and the differences in what we know and what the authors propose quite clear.

I only have some comments to improve this review, in particular on specific aspects of the literature that are not mentioned.

Minor comments:

  • Some repetitions should be checked in particular about the origin of microglia.
  • Infected macrophages in tissues are very often multinucleated (see Orenstein JM. Immunobiology. 2001 or Anderson JM. Curr Opin Hematol. 2000). This is particularly true in the brain. The authors should at least mention MGC (multinucleated giant cells) in their manuscript.
  • The work from the lab of Kuroda MJ should be cited, in particular Cai Y, et al . J Leukoc Biol.
  • Line 28: osteoclasts are not located in the bone marrow but in bones.
  • Line 48: In addition to phagocytosis of infected T cells by macrophages, another mechanism has been demonstrated in vitro by the group of S. Benichou (see the first description in Bracq L, J Virol, 2017). This mechanism of cell-to-cell fusion leading to the formation of heterocaryons containing T cell nuclei is totally in agreement with the in vivo observation of T cell material in infected macrophages by Calantone, Immunity, 2014. These different mechanisms of macrophage infection by cell-to-cell transfer is very well described in a recent review from the group of Q. sattentau (Dupont M, viruses, 2020) that you should be cited here.
  • Line 290: In this paragraph, the work of Melendez LM. on cathepsin B-induced neurotoxicity could be mentioned (for example: Zenón F, J Neurovirol. 2015).
  • Line 310: to the question “are HIV-1 infected macrophages contribute to HIV dissemination ?”, a work published in Blood is partially answering (Vérollet C et al., blood, 2015). A section of the manuscript should be included to discuss the potential mechanisms of HIV-1 dissemination, including migration across the BBB (to reach the brain).

Author Response

We thank the reviewer for the thoughtful comments. We have addressed the concerns, which are detailed in the response below as well as indicated by changes in the manuscript.

Reviewer #1:

This review by H. Akiyama and S. Gummuluru deals with the roles of macrophages during the persistent inflammatory responses upon HIV-1 infection. The manuscript is very interesting as it reviews the available literature in the field to estimate how macrophages could contribute to the anti-viral immune response in tissues, mainly focusing on the brain. I found all the details extremely organized and the differences in what we know and what the authors propose quite clear. I only have some comments to improve this review, in particular on specific aspects of the literature that are not mentioned.

Some repetitions should be checked in particular about the origin of microglia.

We have changed the manuscript to reduce repetitions.

Infected macrophages in tissues are very often multinucleated (see Orenstein JM. Immunobiology. 2001 or Anderson JM. Curr Opin Hematol. 2000). This is particularly true in the brain. The authors should at least mention MGC (multinucleated giant cells) in their manuscript. The work from the lab of Kuroda MJ should be cited, in particular Cai Y, et al . J Leukoc Biol.

The suggested papers have been cited and multinucleated macrophages are mentioned in the revised manuscript (line 33-34).

Line 28: osteoclasts are not located in the bone marrow but in bones.

We have corrected the sentence.

Line 48: In addition to phagocytosis of infected T cells by macrophages, another mechanism has been demonstrated in vitro by the group of S. Benichou (see the first description in Bracq L, J Virol, 2017). This mechanism of cell-to-cell fusion leading to the formation of heterocaryons containing T cell nuclei is totally in agreement with the in vivo observation of T cell material in infected macrophages by Calantone, Immunity, 2014. These different mechanisms of macrophage infection by cell-to-cell transfer is very well described in a recent review from the group of Q. sattentau (Dupont M, viruses, 2020) that you should be cited here.

The role of MDM-T cell fusion is described (line 51-53) and suggested papers have been added to the revised manuscript.

Line 290: In this paragraph, the work of Melendez LM. on cathepsin B-induced neurotoxicity could be mentioned (for example: Zenón F, J Neurovirol. 2015).

We apologize for the confusing statement in Line 290 (in the previous manuscript), which was intended to describe poor phagocytic activity by activated microglia. We have modified the statement. Although we thank the reviewer for their suggestion to include the report, we believe that there are too many reports on microglia-related neurotoxins to cite individually in this review, and we hope these topics would be covered in a review specialized in neuroinflammation.

Line 310: to the question “are HIV-1 infected macrophages contribute to HIV dissemination ?”, a work published in Blood is partially answering (Vérollet C et al., blood, 2015). A section of the manuscript should be included to discuss the potential mechanisms of HIV-1 dissemination, including migration across the BBB (to reach the brain).

We thank for the reviewer’s suggestion. We believe that monocytes (and not macrophages) are thought to be involved in HIV neuroinvasion mechanisms, and discussion of monocyte migration is out of the current review’s scope, which we hope will be discussed in another review article in the future.

Reviewer 2 Report

I thought this is an excellent review article entitled “HIV-1 persistence and chronic induction of innate immune responses in macrophages”. I have gone through the whole article. The literature presented looks very convincing and updated and can be useful for researches in this field. In this manuscript, the authors made a point to discuss the roles of macrophages in facilitating viral persistence and contributing to the generation of persistent inflammatory responses.

A few comments that in my opinion could serve to improve the manuscript:

  1. Typo error should be checked by authors like at line 49 leadss, line 77 HV+ etc.
  2. There is limited discussion of macrophage function and subsets and how these may engage innate immune activation by different mechanisms.
  3. The authors need to discuss more in detail about the mode of HIV-1 persistence and chronic induction mechanism in the innate immune response.

Author Response

We thank the reviewer for the thoughtful comments. We have addressed the concerns, which are detailed in the response below as well as indicated by changes in the manuscript.

Reviewer #2

I thought this is an excellent review article entitled “HIV-1 persistence and chronic induction of innate immune responses in macrophages”. I have gone through the whole article. The literature presented looks very convincing and updated and can be useful for researches in this field. In this manuscript, the authors made a point to discuss the roles of macrophages in facilitating viral persistence and contributing to the generation of persistent inflammatory responses. A few comments that in my opinion could serve to improve the manuscript:

Typo error should be checked by authors like at line 49 leadss, line 77 HV+ etc.

We have corrected these errors in the revised manuscript.

There is limited discussion of macrophage function and subsets and how these may engage innate immune activation by different mechanisms.

Description of macrophage subsets and functional specialization is beyond the scope of the present review. There are other excellent reviews in the literature which have dealt extensively with macrophage ontogeny and functional classification. In response to the reviewer’s query, we have briefly described a potential role of different macrophage phenotypes in pathological consequences (line 307-313).

The authors need to discuss more in detail about the mode of HIV-1 persistence and chronic induction mechanism in the innate immune response.

Latency in tissue-resident macrophages and potential mechanisms of reactivation have been discussed in the revised manuscript (line 90-94).